# Developments in Vascular-Targeted Photodynamic Therapy for Urologic Malignancies

**DOI:** 10.3390/molecules25225417

**Published:** 2020-11-19

**Authors:** Lucas Nogueira, Andrew T. Tracey, Ricardo Alvim, Peter Reisz, Avigdor Scherz, Jonathan A. Coleman, Kwanghee Kim

**Affiliations:** 1Urology Service, Department of Surgery, Memorial Sloan Kettering Cancer Center, New York, NY 10065, USA; nogueirl@mskcc.org (L.N.); traceya@mskcc.org (A.T.T.); ricardoalvim1103@gmail.com (R.A.); reiszp@mskcc.org (P.R.); colemanj@mskcc.org (J.A.C.); 2Department of Plant and Environmental Sciences, The Weizmann Institute of Science, Rehovot 7610001, Israel; avigdor.scherz@weizmann.ac.il; 3Department of Surgery, Memorial Sloan Kettering Cancer Center, New York, NY 10065, USA

**Keywords:** vascular-targeted photodynamic therapy, prostate cancer, urothelial cancer

## Abstract

With improved understanding of cancer biology and technical advancements in non-invasive management of urological malignancies, there is renewed interest in photodynamic therapy (PDT) as a means of focal cancer treatment. The application of PDT has also broadened as a result of development of better-tolerated and more effective photosensitizers. Vascular-targeted PDT (VTP) using padeliporfin, which is a water-soluble chlorophyll derivative, allows for tumor-specific cytotoxicity and has demonstrated efficacy in the management of urologic malignancies. Herein, we describe the evolution of photodynamic therapy in urologic oncology and the role of VTP in emerging treatment paradigms.

## 1. Introduction

With more than two million new cases of prostate, bladder, and kidney cancer diagnosed worldwide annually, malignancy of the urinary tract has a significant impact on global health [1]. Furthermore, the increasing age of populations in developed nations has increased disease burden. While extirpative surgery and traditional radiation therapy have been the mainstay of treatment for control of these diseases at the clinically localized stage, the growing effort to reduce treatment-related morbidity has led researchers to seek less invasive therapeutic options that do not sacrifice oncologic efficacy.

Photodynamic therapy (PDT) has seen renewed interest among urologic oncologists as a cancer treatment modality, attributable, in part, to more powerful and safer photosensitizing agents and more convenient light delivery systems. A growing body of preclinical and clinical evidence suggests that PDT may be an effective treatment for cancers of the prostate, bladder, and kidney, particularly when vascular-targeted photosensitizing agents are utilized. The following narrative review discusses the history of photodynamic therapy in urologic cancers as well as the evidence supporting current and future clinical applications of vascular-targeted photodynamic therapy (VTP).

## 2. PDT and Urological Cancer

Photodynamic therapy is a minimally invasive tissue ablation modality in which a photosensitizing substance is activated through exposure to laser light radiation delivered at a specific wavelength. In the presence of oxygen, this triggers a photochemical reaction that generates oxidant species (radicals, singlet oxygen, triplet species), leading to targeted tissue destruction through direct cytotoxicity, vascular shutdown, and activation of an immune response [2].

Initially described in 1903 for the treatment of skin cancer [3], several agents have been tested and used for PDT in the ablation of various tumors. Most PDT techniques are based on cellular-targeted photochemotherapy (CTP) in which the photosensitizer preferentially accumulates in parenchymal cells, causing local damage through light radiation. Although many agents have been evaluated, only a few have been approved for clinical use. Photosensitizing agents that are currently approved or being tested in clinical trials are listed in Table 1.

There has been growing interest in PDT as a treatment modality for prostate cancer over the past two decades, driven in part by earlier detection due to increased use of prostate imaging. In addition, the functional damage, such as urinary and sexual dysfunction, caused by radical treatments for prostate cancer, including radical prostatectomy and radiation therapy, have favored development of less invasive therapeutic modalities [14]. Addressing this concern, PDT offers comparable oncological efficacy in treating the index lesion, while theoretically sparing the urethra and the neurovascular bundles from debilitating damage. Early studies of PDT for prostate cancer used trans-urethral or trans-perineal irradiation with photosensitizers such as hematoporphyrin derivative meso-tetra-(m-hydroxyphenyl)chlorin, 5-aminolevulinic acid, motexafin lutetium, or temoporfin [15]. Unfortunately, these initial studies were limited by an inability to correctly localize the lesions of interest as well as poor selectivity for tumor tissue, leading to significant side effects. This poor selectivity resulted from the fact that these studies tested CTP strategies, which do not safely preserve sensitive surrounding structures including the neurovascular bundles, rectum, and urethral sphincter. Furthermore, the long half-life of early photosensitizing agents led to prolonged phototoxicity [16].

More recently, critical developments in understanding the disease dynamics, along with image and devices improvements, helped to consolidate the role of PDT in prostate cancer. In this scenario, the consolidation of the index lesion concept and tumor behavior patterns were paramount. The development and wider use of other types of focal therapy as high-intensity focused ultrasound (HIFU) and electroporation contributed to improve PDT techniques. Moreover, the advances in imaging methods including multiparametric magnetic resonance imaging have improved the accuracy of prostatic lesion localization, particularly clinically significant disease [17]. These advances made greater accuracy in the insertion of lighting fibers possible, increasing the effectiveness of PDT-based therapies. Furthermore, the development of photosensitizers such as padoporfin and padeliporfin that accumulate in endothelial cells, called vascular-targeted photodynamic therapy (VTP), causing damage in the vascular environment after light activation, have allowed the development of new focal treatments for prostate cancers, which preserve adjacent structures and yield better functional results.

Upper tract urothelial cancer encompasses neoplasia involving the renal pelvis or the ureter. The natural access throughout the urethra and the well-developed ureteroscopy technics provide the possibility of accessing the tumors without needing insertion with minor aggressiveness. The use of the photosensitizer has helped to improve tumor detection in this setting [17,18,19,20].

## 3. VTP Mechanism of Action

As mentioned above, vascular-targeted photodynamic therapy (VTP) is the latest form of PDT and distinct from CTP. The photosensitizers employed in this treatment, the first generation padoporfin (WTS-09), and the second generation padeliporfin (WST-11), spontaneously and noncovalently forms a complex with serum albumin, which circulates in the blood until cleared, with minimal or no extravasation to adjacent tissues. They generate an intense local release of cytotoxic reactive oxygen species (ROS) upon illumination with the 753-nm laser light. ROS-induced damage to the tumor vasculature results in complete tumor collapse, while preserving organ collagen structure. VTP photosensitizers remain confined within the circulation even at high doses, with minimal, or even no, extravasation to other tissues, and are rapidly cleared by the hepatic and renal systems [21]. Therefore, ROS generated upon laser activation, mostly superoxide and hydroxyl radicals, are similarly contained in the vasculature and do not directly kill tumor cells.

A preclinical study of VTP by Madar-Balakirski et al. revealed that a single illumination led to tumor necrosis with 24–48 h, and eradication and healing several weeks after that [22]. This study also showed the immediate ROS impact is within the tumor circulation and not in vascular endothelial cells via clots in the tumor-feeding arteries, which is followed by the vein system, leading to irreversible vascular collapse and blood stasis. These clots formed on the inner artery walls, particularly at precapillary bifurcation points, a few seconds after initiation of laser illumination, and mobilized toward the interface of the tumor microcirculation, leading to vessel occlusion. This occlusion is thought to result from the unusual fragility and absence of contractile smooth muscle elements in tumor vessels, which makes them selectively susceptible to VTP-induced collapse. In this study, the adjacent healthy tissue remained functional with intact vessels, despite exposure to the same ablative procedure.

Importantly, VTP’s effects are nonthermal. Kimm et al. demonstrated that the biochemical reaction in VTP does not induce heat or thermal injury [23]. At 208 mW/cm^2^ laser energy, VTP did not generate a clinically significant temperature change, defined as ΔT ≥ 8 °C because coagulation effects would be expected with extended animal tissue heating to >45 °C. Together, these studies show that VTP leads to nonthermal, complete tissue ablation characterized by uniform coagulation necrosis, while preserving tissue collagen scaffolding, blood vessels ≥40 μm in diameter, and surrounding stroma. These effects preserve normal organ function and allow fast recovery after treatment.

The Kimm et al. study, performed in non-tumor-bearing swine, also showed that this approach spares vessels ≥40 μm in diameter, confirming its safety in proximity to important large vessels, such as in the ureter and renal pelvis [23]. Finally, the authors determined optimal laser fiber distribution, specifically spaced 1 cm apart, and the relationship between laser fluence and the radius of tissue ablation. A companion study similarly confirmed the safety of endoluminal ablation in the ureter and renal pelvis and the intensity dependence of the depth of tissue necrosis [24]. This investigation further showed that tissue in the irradiated area, in this case, the urothelium, regenerates within 4 weeks with no functional changes. 

## 4. New Imaging Methods to Evaluate a VTP Response in Urologic Cancers

The only imaging method that has been clinically assessed for the monitoring of the VTP response is magnetic resonance imaging (MRI) [25], but studies, so far, have been small series without comprehensive tissue-based correlation [26,27]. The accuracy of post-treatment MRI is inconsistent possibly due to low-volume disease, anatomic distortion, and fibrosis, which can influence sampling accuracy and radiographic interpretation [25]. Toward the development of more accurate methods, a variety of noninvasive imaging modalities have been assessed preclinically for monitoring of VTP-induced changes, which have all proven feasible, even though none has yet advanced to the clinic.

Among the sound-based technologies, Cornelis et al. evaluated the accuracy of contrast-enhanced ultrasound (CEUS) in predicting tumor necrosis after VTP in a murine renal cancer tumor model through radiologic-pathologic comparison [28]. CEUS employs gas-filled microbubbles with high compressibility and resonance that make them useful intravascular contrast agents. The size and shape of the necrotic tumor on CEUS images performed at 24 h post-VTP correlated with those determined by pathology with good inter-observer concordance. While CEUS underestimated the necrotic area on pathology by approximately 5%, the difference was not statistically significant. While the study supports the feasibility of CEUS for VTP response monitoring, its accuracy at earlier time points and in non-hyper-vascular tumors remains to be studied. Another acoustic imaging method used for this purpose is multispectral optoacoustic tomography (MSOT), which offers a view of the entire tumor. Neuschmelting et al. assessed whether MSOT-measured changes in oxygen saturation (SO_2_) indicated VTP-induced necrosis [29]. Not only did areas of decreased SO_2_ correlate with pathology findings, but MSOT allowed monitoring of the progressive decreases in SO_2_ over time, from 60% at 1 h to >70% at 48 h post-treatment. The study also showed that MSOT illumination can be used to activate WST11. Building on this work, Haedicke et al. tested the ability of a higher-resolution acoustic imaging method, known as raster-scanning optoacoustic mesoscopy (RSOM), which measures oxygenation of hemoglobin, to evaluate the short-term and long-term anti-vascular therapeutic effects of VTP [30]. RSOM provided a detailed view of the distinct responses of two xenograft models of bladder cancer, employing the UMUC3 and patient-derived 5637 cell lines. At three days post-VTP, while vasculature in 5637 tumors, known to respond well to VTP, was completely collapsed, UMUC tumors presented with a necrotic center but vasculature regrowth at the periphery. These differences suggest that RSOM measurement of a vascular response may be useful for predicting VTP efficacy.

Two studies have also evaluated cancer-targeted radiotracer-based strategies for VTP-related imaging in prostate cancer. Since positron emission tomography/computed tomography using ^68^Ga-labeled prostate-specific membrane antigen (^68^Ga-PSMA PET/CT) is widely used to localize prostate tumors, Alvim et al. assessed its accuracy in detecting tumor recurrence after VTP in a xenograft model [31]. This approach detected all recurrent tumors and had negative results in all others, supporting its use for follow-up in patients after VTP to detect tumor recurrence early (Figure 1). While not strictly a response assessment, Cerenkov luminescence imaging (CLI) has been used to measure retention of the radiotracer/radiotherapeutic ^90^Y-DOTA-AR and a bombesin/ gastrin-releasing peptide receptor (GRPr) antagonist peptide [32] in combination with VTP. VTP was shown to improve retention of this agent, leading to better tumor control in prostate cancer cell-xenografted mice [33].

## 5. Immune Modulation by VTP and Adjunct Immunotherapy

Since VTP-induced hypoxia and cell death initiates an acute local inflammatory cascade within the target tissue, its effects may also be immune-mediated, and it may enhance the efficacy of immunotherapies. Preise et al. were the first to describe the host immune response after VTP [34]. In mice bearing tumors from xenografted colon and mammary carcinoma cells, they demonstrated that a functional immune system is necessary for an effective treatment response and VTP induces long-lasting systemic antitumor immunity, involving both cellular and humoral components.

Given VTP’s enhancement of antitumor immune responses, O’Shaughnessy et al. evaluated whether VTP could potentiate the efficacy of the programmed cell death protein-1 (PD-1) and the programmed death protein ligand-1 (PD-L1) pathway inhibition using an orthotopic murine model of renal cell carcinoma that develops lung metastases [35]. Combined VTP with PD-1/PD-L1 inhibition, but neither treatment alone, resulted in regression of the tumors, prevention of lung metastases, and prolonged survival. Combination therapy led to increases in the ratios of CD8+ and CD4+ cells relative to regulatory T cells in primary renal tumors and increased T cell infiltration in sites of lung metastasis, suggesting a role for both cytotoxic and memory T cells in the enhanced immune response. Similarly, supporting the clinical investigation of combining immune checkpoint inhibitors with VTP, Corradi et al. showed that the combination of VTP and anti-cytotoxic T-lymphocyte-associated protein 4 (CTLA-4) effectively treated the primary tumors, prevented lung metastasis, and extended survival in a urothelial carcinoma syngeneic mouse model [36]. Moreover, in mice previously treated with VTP or with VTP and anti-CTLA-4, cancer cells injected later did not grow into tumors. These studies provide rationale for clinical trials of combination VTP and immune checkpoint inhibitor therapy in both bladder and upper tract tumors.

Since immunosuppressive myeloid cells have also been implicated in limiting the anti-tumor immune response and, thus, promoting tumor growth, and are recruited to sites of wound healing. Lebdai et al. assessed the effect of VTP on infiltration of these cells [37]. Using a cell line-based prostate cancer xenograft model, the authors demonstrated that VTP induces the recruitment of myeloid-derived suppressor cells (MDSCs) to treated tumors as well as their expression of colony-stimulating factor 1 receptor (CSF1R), which is required for myeloid differentiation, proliferation, and tumor migration. They also showed that the combination of VTP with CSF1R blockade inhibited myeloid infiltration and increased CD8+ T cell infiltration, leading to reduced tumor growth and increased survival. This study, thus, supports clinical investigation of this alternative VTP/immunotherapy combination and suggests that VTP may synergize with a range of immunotherapies.

## 6. VTP and Phosphatidylinositol 3-Kinase (Pi3k) Signaling Pathway

The PI3K cell signaling pathway is frequently altered in most cancer and may play an important role in endothelial cell survival and tumor recurrence after VTP. Kraus et al. performed an in vitro study to evaluate the association of three PI3K inhibitors (BYL719, BKM120, and BEZ235) and Verteporfin-VTP in SV40 endothelial and PC-3 prostate cancer cell. An enhanced VTP response were observed with all the three agents, and BEZ235 was presented as the best combination. These preliminary results support the combination of VTP and PI3K pathway inhibitors [38].

## 7. Clinical Trials in Urologic Cancers

### 7.1. Phase I

The first-in-human study of VTP, using padoporfin/WST-09 [39], specifically focused on its potential side effect of photosensitivity because of its residual uptake and longer clearance rate from the skin [40]. No cutaneous phototoxic effects were observed upon light exposure up to 128 J/cm^2^ 1–3 h after treatment with doses of 2 mg/kg, suggesting that photosensitivity after padoporfin-mediated VTP is negligible after a brief waiting period. The phase I study results reported by Trachtenberg et al. included safety, pharmacokinetics, and hypovascular lesion formation, as detected by contrast enhanced magnetic resonance imaging and transrectal ultrasound guided biopsies [41]. In 24 patients with recurrent prostate cancer after external beam radiation therapy (EBRT), escalating doses of both padoporfin and light, delivered using two 763-nm diode laser fibers, were evaluated. No skin photosensitivity was observed up to 48 h post-VTP, and no residual padoporfin in serum was detected 2 h post-treatment. Hypovascular lesion size was drug dose-dependent and light fluence-dependent with the best results observed at 2 mg/kg and 360 J/cm^2^, respectively, and thresholds for detectable hypovascular lesions were defined as 2 mg/kg VTP dose and 100 J/cm^2^ light fluence. No serious adverse events were observed, and sexual, urinary, and bowel function returned to baseline six months after VTP. Contrast-enhanced MRI at one-week post-VTP was considered a reliable early indicator of response as confirmed by a six-month biopsy.

### 7.2. Phase I/II

A phase I/II trial reported by Gertner et al. evaluated the safety, pharmacokinetics, and efficacy of padoporfin-mediated VTP in 15 patients with recurrent prostate cancer after EBRT [42]. In this trial, doses of padoporfin were escalated from 0.1 to 2 mg/kg at a fixed 100 J/cm^2^ light fluence. VTP treatment was technically feasible and well-tolerated in all patients. Plasma concentration of padoporfin was linearly related to dose, and none was detectable in plasma 2 h after irradiation. Padoporfin was not detected in urine at any time point. Hypovascular lesion formation was detected by MRI in patients receiving 1.0 and 2.0 mg/kg doses. Average radiation penetration was 5.5 mm in the lesions and 3.2 mm in VTP-exposed tissue regions in which lesions were not detectable. The authors postulated that the decrease of radiation penetration in the unresponsive areas was caused by fibrosis and calcification in recurrent tumors.

Azzouzi et al. performed a pooled analysis of data from one phase I/II [43] and two-phase II clinical trials [44] involving 117 men with localized, ISUP grade ≤2 prostate cancer [45]. Patients received a 10-min intravenous infusion of a single dose of 4 mg/kg padeliporfin, activated by a 753 nm light at 200 J/cm^2^. Hemi-ablation was performed in unilateral disease while conservative subtotal ablation techniques were used in the bilateral cases.

VTP treatment was technically feasible in all patients. The 6-month negative biopsy rate was 68.4% in the overall cohort and 80.6% for patients treated by hemi-ablation with light density index (LDI) ≥1 (*n* = 67), and the proportion of patients with MRI-detectable necrosis at seven days was 76.5% and 86.3%, respectively. Adverse events occurred in 82.9% of patients. Most of them were mild or moderate, and were related to the drug in 52.1% of patients, the device in 45.3% of patients, and the technical procedure in 76.9% of patients. The most common adverse events were dysuria, urinary retention, erectile dysfunction, and voiding urgency. At six months’ follow-up, there was a slight improvement in urinary function and a slight deterioration of sexual quality. The authors recently published updated data from the two Phase II trials. After 3.5 years of follow-up time, the tumor free rate in the treated lobe was 75%. Side effect rates remained stable with longer follow-up with most of that had a lower grade (5% and 0% grade III and IV, respectively) [46].

### 7.3. Phase II

Trachtenberg et al. evaluated VTP for whole prostate ablation in 18 patients with recurrent prostate cancer after EBRT [47]. Patients were treated with a fixed dose of 2 mg/kg of padoporfin using up to six 763-nm diode laser fibers for light delivery. Patient-specific escalating light doses and fiber positioning were determined by planning software based on pre-treatment MRI. VTP treatment was technically feasible in all patients. Stronger light doses were associated with better treatment response, encompassing up to 80% of the prostate volume. Complete pathological responses at 6-month biopsy required ≥23 J/cm^2^ light dose in 90% of the prostate and was seen in 8 of 13 patients who received such treatment. MRI changes were observed in all patients, and >60% of image responses were associated with a complete pathological response. Sexual, urinary, and bowel function returned to baseline 6 months after VTP. Rectal wall devascularization was observed in 10 patients, while recto-urethral fistula was diagnosed in two. These results support VTP’s utility for the treatment of prostate cancer recurrence after EBRT.

VTP was first evaluated for the treatment of localized prostate cancer by Azzouzi et al. [48], who investigated the optimal padeliporfin concentration and light dose parameters to achieve prostate cancer ablation in patients who met active surveillance criteria. Patients (*n* = 86) were divided into two groups with one receiving a fixed light dose of 200 J/cm^2^, and the other light doses varying from 200–300 J/cm^2^. Drug concentration was defined by prostate size in both groups: 4 mg/kg if <60 mL, and 6 mg/kg for prostates ≥60 mL. VTP treatment was technically feasible in all but one patient. Optimal treatment parameters were defined as a padeliporfin concentration of 4 mg/kg and 200 J/cm^2^ light, which led to an 83% negative biopsy rate at 6 months and 88% necrosis at 7-day MRI. Adverse events occurred in 87% of patients in which most of them are mild or moderate. At 6 months’ follow-up, urinary function generally improved and sexual quality slightly deteriorated.

One year later, Moore et al. published a similar study, aiming to determine the optimal light and drug doses for VTP using padeliporfin in men with localized low-risk prostate cancer [49]. A total of 42 patients were treated with a fixed light dose of 200 J/cm^2^ using two 753-nm diode laser fibers for delivery following infusion of padeliporfin over 10 min at escalating doses of 2, 4, and 6 mg/kg. VTP treatment was technically feasible in all but two patients. Treatment with the 4 mg/kg drug dose activated by a 753-nm light at 200 J/cm^2^ and an LDI of >1 led to the best outcomes with a treatment effect in 95% of the planned treatment volume and a negative biopsy rate at 6 months of 83%. Adverse events occurred in 81% of patients, which were mostly mild or moderate. At 6 months’ follow-up, there was no significant change in urinary or sexual function relative to baseline.

Another trial evaluated the efficacy of VTP hemiablation using 4 mg/kg padeliporfin in men with low-risk and intermediate-risk prostate cancer, including those with bilateral ISUP group 1 and 2, yielding similar results to earlier phase II studies [50]. At 12 months, 60 of 81 (74%) men had negative biopsies. Of the eight patients with ISUP group 2 prostate cancer who had biopsies at 6 and 12 months, all had negative biopsies at 12 months, and none underwent subsequent prostate cancer treatment during the study period.

Taken together, these studies demonstrate a strong correlation between total light energy delivered and efficacy, and defined the optimal VTP treatment parameters for subsequent phase III studies: 4 mg/kg padeliporfin, 200 J/cm^2^, and LDI ≥1.

### 7.4. Phase III

In 2016, Azzouzi et al. published the only prospective randomized phase III study comparing VTP using padeliporfin versus active surveillance in 413 patients with low-risk prostate cancer. After an average follow-up of two years, patients undergoing VTP had a lower rate of disease progression (28% vs. 58%, hazard ratio(HR): 0.34, 95% confidence interval (CI) 0.24–0.46, *p* < 0.0001), higher rate of negative biopsies (49% vs. 14%, HR: 3.67, 95% CI 2.53–5.33, *p* < 0.0001) and less need for radiotherapy or surgery (6% vs. 29%, *p* < 0.0001) [51]. In the group treated with VTP, patient reports on the IIEF-15 and IPSS questionnaires showed a deterioration of erectile and voiding functions in the first 6 months, but, after two years’ follow-up, these outcomes did not significantly differ when compared with active surveillance (*p* = 0.64). As expected, both the frequency and severity of adverse effects were higher among patients treated with VTP, but most patients had mild or moderate complications (Grade 1 or 2) that occurred in the first few days after the procedure, and fully recovered without sequelae. Perineal pain (15%) and urinary infection (10%) were the most common complications. This study led to the approval of VTP for the treatment of low-risk localized prostate cancer in Europe.

## 8. Current Studies in Prostate and Urothelial Upper Tract Cancers

A single-center single-arm phase IIb study is currently investigating VTP’s efficacy, safety, and effects on quality of life in 50 patients with intermediate-risk prostate cancer [9]. The research includes men with a histologic diagnosis ISUP group 2 disease on one half of the prostate in ≤2 sextants of the prostate gland and not present in >50% of any one core, cT2a disease, prostate volume between 25–70 mL, and serum prostate-specific antigen (PSA) ≤10 ng/mL. Patients will be followed for 5 years by clinical evaluation, PSA, and prostate biopsies at 3, 12, 24, 36, 48, and 60 months. According to an early report, 40 of 49 patients (82%) had no ISUP group 2 or more significant cancer in the index lobe at 3 months [9]. Eleven (22%) underwent per-protocol second hemi-ablation treatment for the ISUP group 2 tumor at 3 months: nine for residual cancer and three for newly identified tumors. Fifteen of the 16 men (94%) who had undergone 12-month biopsy had no Gleason grade 4 or 5 cancer, including 6 of 7 (86%) receiving two treatments. Final results are expected in the near future.

A single-center single-arm phase I dose-finding study (NCT03617003) is currently evaluating 16 patients with recurrent UTUC treated with up to two sessions of endoscopic padeliporfin VTP. Eligibility included residual or recurrent urothelial carcinoma of the ureter or renal pelvis failing prior endoscopic treatment, who are unable or unwilling to undergo surgical management by resection of the involved ureter or kidney. Padeliporfin (4 mg/kg) was infused over 10 min and activated using a diode laser at 753 nm via a flexible ureteroscope. Light dose escalated from 100 mW/cm^2^ up to a maximally tolerated dose of 200 mW/cm^2^. The primary endpoint is the determination of maximally tolerated laser light fluence rate with secondary objectives of treatment efficacy and biomarker identification.

## 9. Conclusions

Vascular-targeted photodynamic therapy using padeliporfin/WST-11 is a safe and well-tolerated treatment for urological malignancies, as demonstrated in a large number of preclinical and clinical studies. It delivers focal tumor damage secondary to destruction of small vascular vessels, triggered by the release of cytotoxic reactive oxygen species upon specific light illumination. Due to its nonthermal mechanism of action, minimal effects on functional outcomes, and support by level 1 evidence from the only randomized controlled trial yet reported for any focal therapy. VTP holds significant promise as the focal therapy of choice over other modalities for patients with urological cancer. Further studies evaluating oncologic efficacy are warranted, given these promising results.

## Figures and Tables

**Figure 1 molecules-25-05417-f001:**
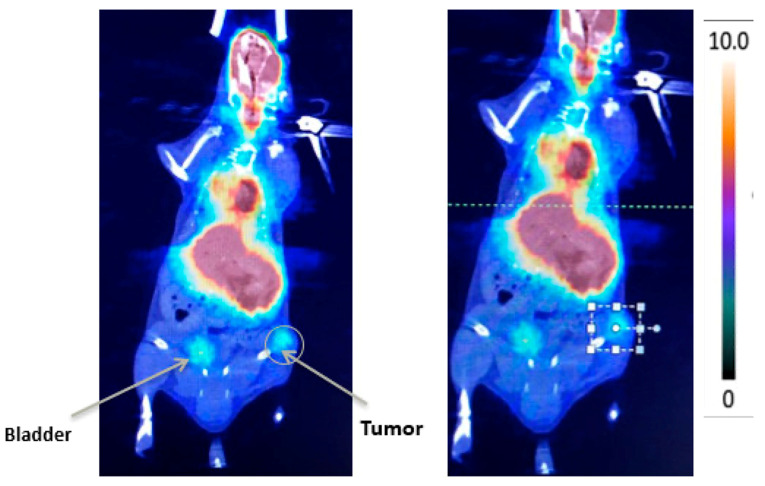
68Ga-PSMA PET/CT images showing tumor recurrence.

**Table 1 molecules-25-05417-t001:** Photosensitizers that have been approved or are under investigation for cancer treatment.

Agent	Cancer Types	Administration
Porfimer sodium [4]	Lung, esophagus, bile duct, bladder, brain, ovarian, breast, skin metastases	Intravenous injection
5-aminolevulinic acid (5-ALA) [5]	Skin, bladder, brain, esophagus	Topical, oral, or intravenous injection
Methyl-aminolevulinate (MAL) [6]	Skin	Topical
Hexyl aminolevulinate (h-ALA) [7]	Skin	Topical
Verteporfin/ benzoporphyrin derivative (BDP) [4]	Pancreas, breast	Intravenous injection
Padeliporfin/ WST-11 [8,9]	Prostate, esophagus, pancreas, urothelial	Intravenous injection
Temoporfin [10]	Head and neck, lung, brain, bileduct, pancreas skin, breast	Intravenous injection
Talaporfin [4]	Liver, colon, brain, lung, breast skin metastases	Intravenous injection
HPPH [11]	Head and neck, esophagus, lung	Intravenous injection
Rostaporfin [4]	Skin, breast	Intravenous injection
Fimaporfin [12]	Skin, bile duct	Intra-tumoral or intravenous injection
Motexafin lutetium [13]	Breast	Intravenous injection

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
