# Peer review of "Developments in Vascular-Targeted Photodynamic Therapy for Urologic Malignancies"

_molecules, 2020, doi:10.3390/molecules25225417_

Round 1

Reviewer 1 Report

The authors in work entitled: “Developments in Vascular-Targeted Photodynamic Therapy for Urologic Malignancies“ summarized up to date information about VTP and its application in urologic cancers. The review consists of scientific studies and clinical trials. I have few comments:

  1. Table 1 consists photosensitizers and their suitability for cancer type and form of administration. It will be more clear if the references will be a part of the table. Like this, the reader can find easily the information.
  2. Page 3/75-84 the section should be supported by references.
  3. Some shortcuts are not introduced, please check the manuscript and correct it. (e.g. PD-1/PD-L1, anti-CTLA-4, HR, CI)

Author Response

November 9th, 2020

Dr. Carlos J. P. Monteiro, Prof. Dr. M. Amparo F. Faustino and Dr. Catarina I. V. Ramos

Guest Editors

Molecules

Dear Editors:

Thank you and the reviewers again for considering our paper for publication in Molecules. All changes in the manuscript are highlighted in the text.

Please find below our comments regarding the REVIEWER 1 points:

  • Table 1 consists of photosensitizers and their suitability for cancer type and form of administration. It will be clearer if the references will be a part of the table. Like this, the reader can find easily the information.

Response: We would like to thank you for pointing this issue out. References were included in the table for each photosensitizer.

  • Page 3/75-84 the section should be supported by references.

Response: We would like to thank you for pointing this issue out. The reference was included in this paragraph.

  • Some shortcuts are not introduced, please check the manuscript and correct it. (e.g. PD-1/PD-L1, anti-CTLA-4, HR, CI)

Response: We would like to thank you for pointing this issue out. The text was amended adequately.

Sincerely,

Lucas Nogueira, MD

Urology Service, Department of Surgery

Memorial Sloan Kettering Cancer Center

Reviewer 2 Report

The paper is very interesting and sound. It has been prepared generally properly but it needs some minor revisions regarding list of references, as the style of presenting numbers of pages of particular references is not uniform. Moreover in reference 27 some bibliographic data are lacking (number of volume and issue, as well as number of first and last page).

The manuscript describes in details the history of application of photodynamic therapy in the treatment of prostate cancer, the mechanisms of vascular targeted photodynamic therapy with the use of novel photosensitizer padeliporfin (a water-soluble chlorophyll derivative) in urologic applications, new imaging methods applied for estimation the efficacy of vascular-targeted photodynamic therapy in urologic malignancies including among others: magnetic resonance imaging, contrast-enhanced ultrasound, multispectral optoacoustic tomography, raster-scanning optoacoustic mesoscopy and positron emission tomography/computed tomography using 68Ga, as well as Cerenkov luminescence imaging, the mechanisms of immune modification by vascular-targeted photodynamic therapy and immunotherapy and, what is especially important the results of clinical trials (phase I, Phase I/II, phase II, phase III) regarding the application of vascular targeted photodynamic therapy in the treatment of urologic malignancies – among others in prostate and urothelial upper tract cancers. Basing on the results of presented preclinical and clinical studies the authors concluded that vascular-targeted photodynamic therapy is safe, well tolerated and efficient method of treatment of urologic malignancies and that after performing further controlled randomized trials it could be treated as potential method of choice in focal therapy of urological cancers.

In my opinion the strengths of the manuscript is especially presenting the results of most of available results of clinical studies confirming the efficacy of vascular–targeted photodynamic therapy in the treatment of urologic malignancies with relation to the clinical phase of particular trials, as well as the proper selection of a great number of actual, reliable references regarding this important medical problem. I did not find any significant weakness of the manuscript, except of lack of uniform style of presenting bibliographic data in particular references composing reference list (it refers to form of presenting last pages of articles, and lacking numbers of volume, issue as well as first and last page in reference 27).

Author Response

November 9th, 2020

Dr. Carlos J. P. Monteiro, Prof. Dr. M. Amparo F. Faustino and Dr. Catarina I. V. Ramos

Guest Editors

Molecules

Dear Editors:

Thank you and the reviewers again for considering our paper for publication in Molecules. All changes in the manuscript are highlighted in the text.

Please find below our comments regarding the REVIEWER 2 points:

We would like to thank you for the comments in your review. Regarding the reference formatting, we apologize for that, and it was adjusted by the EndNote software in order to format them according to the journal’s rules. About the specific reference #27 (now ref#31), although a 2019 paper, it is still waiting for the final print reference by Eur Urol Focus.

Sincerely,

Lucas Nogueira, MD

Urology Service, Department of Surgery

Memorial Sloan Kettering Cancer Center